# Seeing Beyond Points: Adaptive Gaussian Primitives for 3D Perception

## Abstract

The sparse and discrete nature of point clouds fundamentally limits their effectiveness in perception tasks, as these raw 3D data collections inadequately capture the continuous geometry and detailed appearance of complex real-world scenes. We propose **GCept**, a unified 3D perception framework that evolves raw points into adaptive Gaussian primitives, representing a natural progression in point cloud enrichment. GCept groups spatially proximate points into 3D Gaussians with optimized covariances and spherical harmonics encoding, forming a continuous density field that preserves intricate geometric structures and subtle visual details often lost in traditional pipelines. To enhance representational quality, GCept employs an alpha-guided sampling mechanism that strategically uses compositing weights from Gaussian Splatting to retain only the most informative primitives. The resulting enriched Gaussian representation integrates seamlessly into standard 3D perception backbones, providing richer geometric and appearance information for downstream tasks. Experiments on ScanNet, ScanNet++, ScanNet200, and S3DIS demonstrate state-of-the-art performance in semantic and instance segmentation, effectively bridging 3D reconstruction with robust perception.

## 1 Introduction

Point clouds, the direct and unstructured output from 3D sensors, represent the world as millions of spatial samples. However, these samples inherently lack continuity and adjacency and frequently exhibit non-uniform density, presenting substantial challenges for direct utilization. Historically, both rendering and perception pipelines have endeavored to enrich this raw data. For instance, as illustrated in Figure 1(A), rendering techniques progressed from simple points to Surfels Pfister et al. (2000), oriented discs incorporating radius, normal, and color. Analogously, perception pipelines augmented accuracy by supplementing points with pre-computed normals Qi et al. (2017a;b); Choy et al. (2019); Graham et al. (2018). The latest advancement along this trajectory is 3D Gaussian Splatting (3DGS) Kerbl et al. (2023), which models local point clusters as anisotropic Gaussians, complete with mean, covariance, and spherical harmonics for view-dependent color, thereby facilitating photorealistic rendering.

This paper argues for rethinking the point cloud's intension to broaden its extension. We view 3D Gaussians not as a substitute for points, but as the next logical step in their evolution, following the lineage from raw coordinates to Surfels and attribute-augmented points. Just as incorporating normals enhances perception, adopting Gaussian primitives transforms each discrete sample into a localized continuous model capturing geometry, uncertainty, and subtle appearances frequently omitted by sparse points, equipping each sample with richer descriptive power for 3D perception.

To accomplish this objective, we propose GCept, which exploits Gaussian Splatting for 3D perception. As depicted in Figure 1(B), GCept aggregates spatially proximate points into 3D Gaussians characterized by spatial mean, orientation, and scale. This yields a compact density field that encapsulates geometric structures and encodes view-dependent appearance through spherical harmonics, preserving visual details often lost in conventional representations.

While Gaussian fields are richer than a bare point sets, naively retaining every Gaussian introduces redundancy. Existing pipelines down-sample by uniformly selecting one point per grid cell Choy et al. (2019); Wu et al. (2024), potentially discarding critical regions like thin structures and object boundaries. We consequently introduce an alpha-guided reweighted sampling mechanism that ex-

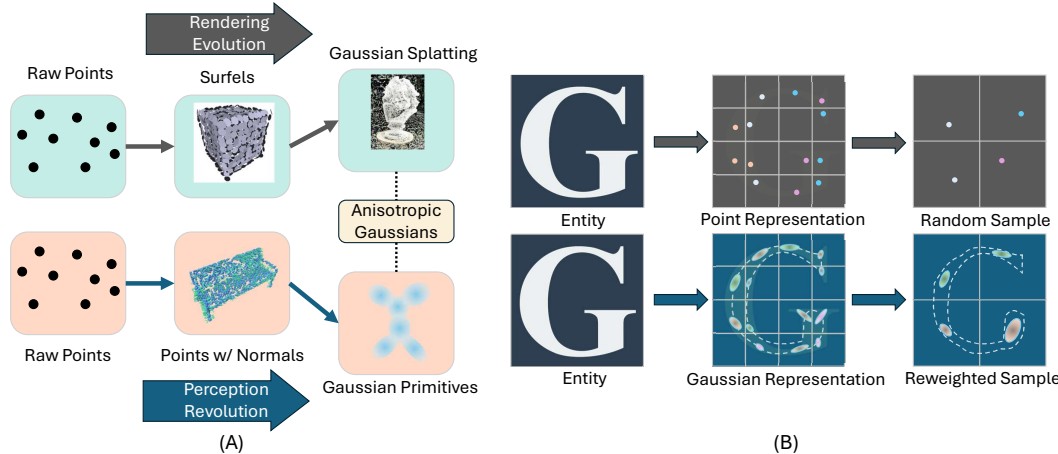

Figure 1: (A) Parallel evolution of rendering and perception, both progress from raw points to anisotropic Gaussians, underscoring Gaussians as the shared, continuous representation. (B) Given the same sensor data, uniform/random sampling of discrete points eliminates thin structures, whereas GCept forms Gaussian primitives and preserves them via alpha-guided reweighted sampling.

ploits the compositing weights generated by Gaussian Splatting. During rasterization, each Gaussian contributes an alpha term that quantifies its visibility along every camera ray. By aggregating these terms across multiple views, we derive an importance score that reflects the degree to which each Gaussian influences the scene's appearance. Sampling Gaussians proportional to this score concentrates network capacity on geometrically and visually informative regions, diminishes ambiguity at object boundaries, and mitigates class imbalance without additional supervision.

Building upon these two efficacious designs (*i.e.*, Gaussian-based representation and adaptive reweighted sampling), GCept attains substantial improvements over conventional point-based representations on standard indoor benchmarks. Furthermore, comprehensive ablation studies demonstrate that our Gaussian-based approach effectively diminishes ambiguity at object boundaries and enhances the delineation of fine-scale details in complex scenes. In summary, our contributions are as follows:

- **Gaussian-based representation as point cloud evolution.** We leverage Gaussian Splatting not merely as a reconstruction instrument but as a methodology to enhance the fundamental representation of point cloud data for perception, preserving richer geometric and appearance information directly within each primitive.

- **Adaptive reweighted sampling.** We propose a novel sampling strategy that prioritizes the most informative Gaussians based on their rendering contribution (e.g., alpha compositing weights from Gaussian Splatting), effectively reducing redundancy while maintaining critical details that are necessary for accurate perception.

- **Demonstrated performance gains.** We demonstrate through extensive experiments on benchmarks like ScanNet, ScanNet++, ScanNet200, and S3DIS Dai et al. (2017); Yeshwanth et al. (2023); Armeni et al. (2016) that integrating our GCept framework into standard 3D perception backbones yields significant improvements in semantic and instance segmentation tasks compared to traditional point-based approaches.

## 2 RELATED WORK

**3D understanding.** 3D deep-learning pipelines can be grouped into four families: projection-based Su et al. (2015); Chen et al. (2017), voxel-based Maturana & Scherer (2015); Zhou & Tuzel (2018); Thomas et al. (2019), point-based Qi et al. (2017a;b); **?**); Lai et al. (2022); Qian et al. (2022); **?**, and multi-modal 2D–3D fusion methods Hu et al. (2021); Jain et al. (2024). While projection methods may lose depth information and voxel methods face memory constraints, point-based networks with transformers achieve strong performance but struggle with long-range dependencies. Multi-modal

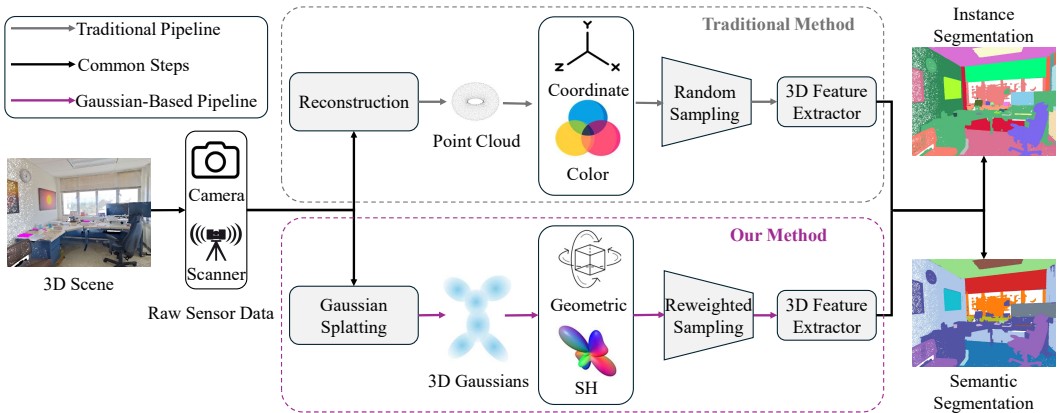

Figure 2: Overview of our pipeline. We illustrate two distinct approaches to processing raw sensor data from cameras and scanners. In the traditional pipeline (gray arrows), a reconstruction step produces a point cloud with coordinate and color attributes, which then undergoes random sampling before entering a 3D feature extractor. In contrast, our Gaussian-based pipeline (magenta arrows) applies Gaussian splatting directly to the raw sensor data, creating a set of 3D Gaussians enriched with geometric and spherical harmonic (SH) features. A reweighted sampling strategy then preserves the most informative Gaussians. Both pipelines feed into the same 3D backbone for downstream perception tasks, such as semantic and instance segmentation.

fusion achieves high accuracy at the cost of additional imaging requirements and pretrained 2D backbones.

**3D Gaussian Splatting.** 3D Gaussian Splatting represents scenes using anisotropic Gaussian kernels, providing a compact alternative to traditional methods Kerbl et al. (2023); Yang et al. (2024). It generates continuous density fields for differentiable rendering and view synthesis. Moreover, 2D Gaussian Splatting (2DGS) Huang et al. (2024) projects the 3D volume into 2D Gaussian disks, achieving view-consistent geometry and real-time performance.

**Integrating 3D reconstruction and perception tasks.** Recent work bridges 3D reconstruction and semantic perception. While traditional methods like SfM Snavely et al. (2006); Schonberger & Frahm (2016) provide accurate geometry without semantics, modern approaches integrate segmentation into reconstruction Dai et al. (2018); Sun et al. (2021); Rosinol et al. (2020). Neural representations like NeRF Mildenhall et al. (2020) and PeRFception Jeong et al. (2022) demonstrate unified reconstruction and perception for holistic scene understanding.

## 3 METHOD

### 3.1 THE GCEPT PIPELINE

As depicted in Figure 2, GCept aggregates spatially proximate points into 3D Gaussians with continuous attributes encapsulating geometry and color (Section 3.2). The scale and rotation matrices encode spatial extent and orientation, while spherical harmonic coefficients provide view-dependent color variations. GCept fixes each Gaussian's center during Gaussian Splatting, maintaining alignment with the original point cloud. A weighted grid sampling strategy (Section 3.3) then uses alpha compositing weights to bias toward the most salient Gaussians, efficiently reducing data density while preserving essential structural and appearance characteristics.

Subsequently, the resultant Gaussian-based representation is seamlessly integrated into a point cloud backbone. In our experiments, we employ both Point Transformer v3 (PTv3) Wu et al. (2024) and SparseUnet Choy et al. (2019); Contributors (2022) to substantiate that GCept can be incorporated into diverse 3D architectural frameworks. By synthesizing GCept's continuous, view-aware features with the versatility of these backbones, the pipeline attains robust performance on tasks including semantic and instance segmentation. Next, we present the details of GCept.

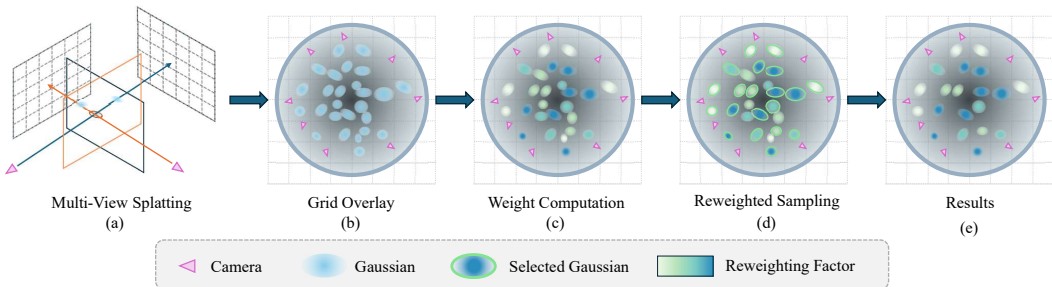

Figure 3: Overview of the weighted grid sampling. (a) Multi-View Splatting: Gaussians are projected onto different camera viewpoints. (b) Overlay the Grid: A uniform grid is applied to the point cloud. (c) Weight Computation: Each Gaussian is assigned a reweighted factor based on alpha compositing. (d) Reweighted Sampling: Probabilistic selection is applied to retain the most informative Gaussians. (e) Result: The final subset of Gaussians represents the scene with reduced redundancy while maintaining critical structure and appearance.

## 3.2 GAUSSIAN-BASED POINT CLOUD REPRESENTATION

To enhance the expressiveness and efficiency of point cloud processing, we introduce a Gaussian-based representation that supplants individual points with spatially extended distributions. Rather than treating each point as an isolated entity, we group spatially proximate points and represent each cluster as a 3D Gaussian. This formulation yields a more structured description of the underlying geometry and appearance, smoothing the discrete nature of raw point clouds while improving representational efficiency and preserving requisite spatial fidelity.

**Gaussian representation and parameters.** Given a point cloud $\mathcal{P} = \{p_i\}_{i=1}^{N}$, we decompose it into smaller clusters $\{\mathcal{G}_j\}_{j=1}^{M}$ comprising spatially adjacent points, where $M \ll N$. Here, $N$ and $M$ denote the cardinality of the point cloud and the partitioned clusters, respectively. Each cluster $\mathcal{G}_j$ is conceptualized as a Gaussian distribution parameterized by: 1) Mean $\mu_j = \frac{1}{|\mathcal{G}_j|} \sum_{p_i \in \mathcal{G}_j} [x_i, y_i, z_i]^T$, denoting the centroid; 2) Covariance matrix $\Sigma_j = R_j S_j^2 R_j^T$, where $S_j$ quantifies scaling and $R_j$ stipulates orientation. This approach conserves geometric structure while substantially mitigating redundancy, facilitating a more structured representation of local geometry within the point cloud.

**Encoding color with spherical harmonic coefficients.** To incorporate appearance information, we adopt spherical harmonic coefficients in accordance with the Gaussian Splatting paradigm, which yields a compact and expressive encoding of view-dependent color variations. This representation enables each Gaussian to capture complex, view-dependent appearance variations beyond rudimentary per-point RGB values by encoding color as a function of viewing direction relative to the local coordinate system of each Gaussian.

**Constructing the Gaussian-based point cloud.** With each cluster represented as a Gaussian, the complete point cloud representation is formally articulated as:

$$\mathcal{S}(\mathbf{x}) = \sum_{j=1}^{M} \exp\left( -\frac{1}{2}(\mathbf{x} - \boldsymbol{\mu}_j)^T \Sigma_j^{-1} (\mathbf{x} - \boldsymbol{\mu}_j) \right), \tag{1}$$

By substituting discrete points with Gaussians, this representation encapsulates both geometric and appearance information in a structured, continuous manner. This formulation facilitates more efficient processing while preserving the underlying spatial relationships that are imperative for downstream 3D perception tasks such as semantic segmentation and instance segmentation.

## 3.3 WEIGHTED GRID SAMPLING MECHANISM

As illustrated in Figure 3, our sampling mechanism incorporates a reweighting factor into standard grid-based subsampling, ensuring that the most informative points are preserved. This factor is computed utilizing alpha composition as delineated below.

**Alpha compositing weight.** The rendered alpha compositing weight quantifies the contribution of each 3D Gaussian to the final rendered image. Given that the rasterization of Gaussian Splatting employs alpha blending to accumulate color and transparency in screen space, we exploit this blending process to ascertain the significance of each Gaussian. In Gaussian Splatting, each Gaussian is characterized by its opacity ($\alpha$), RGB color ($C$), and projected 2D position in the rendered image. The final pixel color, $C_{\text{final}}$, is derived through alpha compositing:

$$C_{\text{final}} = \sum_{i}^{N} w_i C_i, \tag{2}$$

where $w_i$ signifies the contribution of the $i$-th Gaussian to the final rendered image, and $N$ represents the total number of Gaussians along the viewing ray for that pixel. Each Gaussian's contribution adheres to the alpha compositing rule:

$$w_i = \alpha_i \prod_{j=1}^{i-1}(1 - \alpha_j). \tag{3}$$

Here, Gaussians are indexed in front-to-back order along the viewing ray from the camera to the pixel, with smaller indices ($j < i$) corresponding to Gaussians proximal to the camera. This formulation guarantees that the foremost Gaussian along a viewing ray exerts the greatest influence, as it appears in front; subsequent Gaussians become progressively occluded, with their contribution diminished by the accumulated transparency of the preceding ones. The resultant compositing weight is contingent upon both the Gaussian's opacity and the visibility of antecedent Gaussians. To derive a global importance measure for each Gaussian, we aggregate its compositing weights across all rendered pixels in the image. This aggregated weight functions as an importance metric, prioritizing Gaussians that contribute more substantially to the visual appearance of the scene.

**Sampling algorithm.** We partition the 3D space into uniform grids (with cell size $g$), consolidate the points that reside within each cell, and assign each point its reweighted factor. Rather than arbitrarily selecting a point from the cell, we implement probabilistic sampling proportional to the reweighted factor. This ensures that points that significantly contribute to the appearance are retained with elevated probability. The selected points subsequently convey all pertinent attributes (3D position, normals, color, and spherical harmonic coefficients) into subsequent processing stages. By concentrating on the most geometrically and visually significant points, our weighted grid sampling preserves pivotal scene details, thereby enhancing performance in downstream tasks such as semantic and instance segmentation.

## 4 EXPERIMENTS

### 4.1 DATASETS AND EVALUATION METRICS

To evaluate our proposed pipeline, we conduct experiments on four widely used indoor 3D point cloud benchmarks: ScanNet Dai et al. (2017), ScanNet++Yeshwanth et al. (2023), ScanNet200, and S3DISArmeni et al. (2016). For each dataset, we follow the standard train/val split provided by the dataset creators. Model performance is evaluated using mean Intersection-over-Union (mIoU) for semantic segmentation and mean Average Precision (mAP) for instance segmentation on the validation set. Additionally, we report per-class IoU to assess the model's effectiveness across various categories in our supplemental materials.

### 4.2 IMPLEMENTATION DETAILS

In this section, we describe the specifics of our implementation, including Gaussian-based point cloud representation and training configuration.

**Gaussian-based point cloud representation.** For each dataset, we preprocess the raw point clouds by grouping points within the same grid, where the grid size determines spatial proximity. Gaussian parameters are then initialized for each group, as described in Section 3.2. Each Gaussian $\mathcal{G}_j$ is initialized with its mean $\mu_j$, a scale matrix $S_j$, and a rotation matrix $R_j$, computed based on the statistical properties of the points within the grid. We implement our method using 2D Gaussian

Table 1: **Semantic segmentation results on four indoor datasets.** We report the mIoU on the validation set for ScanNet, ScanNet++, ScanNet200, and S3DIS. Best results are highlighted in **bold**.

| ScanNet Dai et al. (2017) | | | | | S3DIS Armeni et al. (2016) (Area5) | | |
|---|---|---|---|---|---|---|---|
| **Method** | **Input** | **OA** | **mAcc** | **mIoU** | **Method** | **Input** | **mIoU** |
| SpConv | Voxel | 89.3 | 78.7 | 69.3 | MinkUNet | Voxel | 65.4 |
| MinkUNet | Voxel | 90.6 | 80.4 | 72.2 | ST | Point | 72.0 |
| ST | Point | 91.3 | 83.3 | 74.3 | PointNeXt | Point | 70.5 |
| PointNeXt | Point | 90.0 | 82.0 | 71.5 | OctFormer | Point | 72.5 |
| OctFormer | Point | 91.8 | 83.2 | 75.7 | Swin3D | Point | 72.5 |
| Swin3D | Point | 92.2 | 86.3 | 76.4 | PTv1 | Point | 70.4 |
| PTv1 | Point | 87.9 | 78.5 | 70.6 | PTv2 | Point | 71.6 |
| PTv2 | Point | 91.5 | 84.5 | 75.4 | PTv3 | Point | 73.4 |
| PTv3 | Point | 91.8 | 85.3 | 77.5 | | | |
| GCept (Ours) | Gaussian | **92.5** | **86.4** | **79.0** | GCept (Ours) | Gaussian | **75.1** |
| ScanNet++ Yeshwanth et al. (2023) | | | | | ScanNet200 Rozenberszki et al. (2022) | | |
| **Method** | **Input** | **OA** | **mAcc** | **mIoU** | **Method** | **Input** | **mIoU** |
| SpConv | Voxel | 86.2 | 48.8 | 34.3 | MinkUNet | Voxel | 25.0 |
| MinkUNet | Voxel | 87.2 | 49.3 | 35.8 | OctFormer | Point | 32.6 |
| PointNet++ | Point | - | - | 19.8 | PTv1 | Point | 27.8 |
| PTv2 | Point | 88.5 | 54.4 | 40.7 | PTv2 | Point | 30.2 |
| PTv3 | Point | 88.7 | 54.7 | 42.6 | PTv3 | Point | 35.2 |
| GCept (Ours) | Gaussian | **89.4** | **56.8** | **45.1** | GCept (Ours) | Gaussian | **37.1** |

splattingHuang et al. (2024), which serves as an improved implementation over traditional 3D Gaussian splatting. Spherical harmonics are subsequently employed to encode view-dependent color information, capturing detailed color variations from different perspectives. The feature encoding augments the spatial and color information, enhancing the model's ability to accurately segment and distinguish between object instances. In addition, during the Gaussian Splatting process, we obtain Gaussian attributes and compute the importance weight for each Gaussian based on its alpha compositing contribution. This reweighted factor forms the basis of our weighted grid sampling mechanism, which is applied later in the pipeline to selectively retain the most informative Gaussians while reducing redundancy. Finally, we also include a comparison of different Gaussian Splatting implementations in our supplemental materials, demonstrating that our approach provides a unified framework that can accommodate various Gaussian Splatting instances while maintaining consistent performance benefits.

**Training configuration.** Our pipeline employs a flexible backbone network to process the Gaussian-based point cloud with Gaussian and spherical harmonic features Contributors (2023). In our experiments, we demonstrate the effectiveness of our method using both Point Transformer v3 (PTv3) and SparseUNet, highlighting its adaptability to different architectures. The network is trained for both semantic and instance segmentation tasks across the ScanNet, ScanNet++, and S3DIS datasets. To maintain alignment with the original point cloud, we set the learning rate for the Gaussian center positions $\mu_j$ to zero, thereby fixing the centers of each Gaussian to their initialized positions. All experiments are conducted on 4 NVIDIA L40S GPUs with 48GB of memory. Training each model on the augmented point cloud representation for semantic and instance segmentation tasks takes approximately 48 hours for ScanNet++, 40 hours for ScanNet++, and 28 hours for S3DIS. The batch size is set to 12.

**Computational costs.** We also profile and compare the computational costs associated with our method versus traditional point cloud approaches. A detailed analysis of preprocessing time, training duration, and inference speed is provided in our supplemental materials.

### 4.3 MAIN RESULTS

**Semantic segmentation results.**   Table 1 presents the quantitative performance of our method compared to various voxel-based and point-based approaches on four indoor benchmarks: ScanNet, ScanNet++, ScanNet200, and S3DIS. For ScanNet and ScanNet++, we report the mIoU along with overall accuracy (OA) and mean accuracy (mAcc). Notably, our GCept consistently achieves the highest mIoU scores, with improvements of 2.5% on ScanNet++, 1.9% on ScanNet200, 1.5% on ScanNet, and 1.7% on S3DIS compared to previous state-of-the-art methods. Besides, Table 2 isolates approaches Robert et al. (2022); Hu et al. (2021); Yang et al. (2023); Kundu et al. (2020); Jain et al. (2024) that explicitly fuse RGB imagery with point cloud geometry. Despite this additional modality, GCept surpasses the strongest prior fusion model (ODIN) by +1.2 mIoU and outperforms other 2D–3D baselines by 2.6–8.0 points while discarding the entire 2D branch and any pretrained backbones. This demonstrates that our Gaussian-based representation effectively captures both geometric and appearance information from the scene, eliminating the need for separate pretrained image encoders while delivering superior performance. We also provide qualitative visualization results in our supplemental materials.

**Instance segmentation results.**   The results in Table 4 demonstrate that our approach effectively differentiates object instances in complex indoor environments. Our evaluation on ScanNet v2Dai et al. (2017) and ScanNet++Rozenberszki et al. (2022) benchmarks shows substantial improvements in instance segmentation performance across diverse scene types. For a fair comparison, we adopt a standardized instance segmentation framework based on PointGroup Jiang et al. (2020) across all experiments, with the only variable being the backbone architecture. This comparison highlights how the fixed-center Gaussian representation combined with our PTv3 backbone enhances the model's ability to identify distinct object instances, even in cluttered and challenging indoor scenes with significant occlusions and varying object scales.

**Data efficiency evaluation.**   We evaluate the performance of GCept on the ScanNet data-efficient benchmark Hou et al. (2021). This benchmark challenges models under constrained conditions by limiting either the percentage of available reconstructions (scenes) or the number of annotated points. As shown in Table 5, GCept consistently outperforms previous methods across all these settings, demonstrating superior data efficiency in low-data regimes.

### 4.4 ABLATION STUDIES

**Ablation on Gaussian-based representation.**   Table 6 evaluates the contributions of different features and sampling strategies on ScanNet and ScanNet++. Starting from coordinate-only information (Setting I), we progressively add color (II), normals (III), Gaussian scale and rotation(IV), and spherical harmonics (V), all using random sampling. Finally, we replace random sampling with our reweighted approach (VI). Results show that normals provide substantial geometric cues, while Gaussian scale and rotation enhance spatial modeling by capturing continuous extent beyond discrete points. Adding spherical harmonics improves performance by encoding view-dependent color information not present in simple RGB. Our reweighted sampling strategy further boosts accuracy by prioritizing Gaussians with stronger alpha compositing contributions. Together, these components significantly outperform baseline point-based approaches, confirming the effectiveness of our comprehensive Gaussian representation.

**Ablation on grid size and spatial sampling density.**   We explore the effect of grid size on our reweighted sampling mechanism. As shown in Table 6, increasing the grid size leads to a gradual decrease in performance. Larger grid sizes reduce the number of Gaussian primitives fed to the network, which simplifies the representation but also results in the loss of finer geometric and appearance details. However, when comprehensive Gaussian features such as scale, rotation, and spherical harmonics are included, the performance degradation is less pronounced. This observation indicates that our rich Gaussian representation effectively captures and encodes additional spatial and color information, thereby compensating for the reduced spatial sampling density and maintaining higher segmentation accuracy even at coarser grid resolutions. Notably, our method with 0.04m grid spacing (fewer input primitives) achieves comparable or better performance than point-based methods with denser 0.02m sampling, demonstrating superior representational efficiency. This

Table 2: **Comparison with multi-modal methods on ScanNet.** We compare GCept with methods that utilize 2D image inputs. While these methods rely on pretrained 2D backbones, our approach achieves superior performance without requiring pretrained image encoders.

| Method | | ScanNet | |
| --- | --- | --- | --- |
| | Image Input | Pretrained 2D backbone | mIoU |
| DVA | ✓ | ✓ | 71.0 |
| BPNet | ✓ | ✓ | 73.9 |
| DMFNet | ✓ | ✓ | 75.6 |
| VMF | ✓ | ✓ | 76.4 |
| ODIN | ✓ | ✓ | 77.8 |
| GCept (Ours) | ✓ | ✗ | **79.0** |

Table 3: **Ablation on the degree of spherical harmonics.** We evaluate the effect of varying the maximum order of spherical harmonics on segmentation performance. The table reports mIoU scores on ScanNet Dai et al. (2017) and ScanNet++ Yeshwanth et al. (2023) for grid sizes of 0.04 m and 0.02 m using different degrees of spherical harmonics.

| grid size | ScanNet | | | | ScanNet++ | | | |
| --- | --- | --- | --- | --- | --- | --- | --- | --- |
| | sh=0 | sh=1 | sh=2 | sh=3 | sh=0 | sh=1 | sh=2 | sh=3 |
| grid=0.04 | 78.2 | 78.7 | 77.2 | 77.7 | 43.6 | 44.7 | 44.6 | 44.7 |
| grid=0.02 | 78.8 | **79.0** | 78.3 | 77.9 | 44.2 | **45.1** | 44.9 | 45.1 |

Table 4: **Instance segmentation performance on ScanNet and ScanNet++.** We report mean average precision at different thresholds (mAP25, mAP50) as well as overall mAP.

| Ins. Seg. | ScanNet | | | ScanNet++ | | |
| --- | --- | --- | --- | --- | --- | --- |
| PointGroup | $mAP_{25}$ | $mAP_{50}$ | mAP | $mAP_{25}$ | $mAP_{50}$ | mAP |
| MinkUNet | 72.8 | 56.9 | 36.0 | - | - | - |
| PTv2 | 76.3 | 60.0 | 38.3 | - | - | - |
| PTv3 | 77.5 | 61.7 | 40.9 | 38.1 | 30.0 | 20.1 |
| GCept (Ours) | **78.6** | **62.9** | **41.4** | **42.2** | **35.0** | **23.2** |

Table 5: **Data efficiency evaluation on ScanNet.** This table reports the mIoU scores of different methods under two constraints: limited reconstructions and limited annotations.

| Data Efficient | Limited Reconstruction | | | | Limited Annotation | | | |
| --- | --- | --- | --- | --- | --- | --- | --- | --- |
| Methods | 1% | 5% | 10% | 20% | 20 | 50 | 100 | 200 |
| MinkUNet | 26.0 | 47.8 | 56.7 | 62.9 | 41.9 | 53.9 | 62.2 | 65.5 |
| PTv2 | 24.8 | 48.1 | 59.8 | 66.3 | 58.4 | 66.1 | 70.3 | 71.2 |
| PTv3 | 25.8 | 48.9 | 61.0 | 67.0 | 60.1 | 67.9 | 71.4 | 72.7 |
| GCept (Ours) | **30.9** | **52.2** | **62.5** | **67.8** | **63.9** | **69.6** | **74.0** | **74.8** |

ablation confirms the beneficial properties of our Gaussian-based representation and reweighted sampling mechanism.

**Degrees of spherical harmonics.** We evaluate the impact of the spherical harmonics degree on segmentation performance. Here, the degree indicates the maximum order of harmonic coefficients used to encode view-dependent color variations, and for each color channel, the number of coefficients is given by $(sh + 1)^2$. We conducted experiments with degrees $sh = 0$, $sh = 1$, $sh = 2$, and $sh = 3$ using grid sizes of 0.02 m and 0.04 m, as reported in Table 3. Degree-1 consistently achieves the best performance across both grid sizes, while degree-2 and degree-3 add parameters yet fail to improve and sometimes degrade segmentation quality.

This result diverges from Gaussian Splatting view-synthesis practice (where degree-2 with 9 coefficients per color channel typically performs best), and we attribute it to two task-specific factors: (1) Objective mismatch - View synthesis optimizes per-pixel photometric error and benefits from high-frequency view-dependent color terms (capturing specular lobes, Fresnel effects). Semantic segmentation, however, seeks view-invariant features that delineate class boundaries. Extra SH coefficients mostly model lighting variation irrelevant to semantics, diverting capacity away from geometry cues. (2) Storage economy - While the parameter count only grows slightly (just one projection layer is affected), the storage requirements for the ScanNet++ dataset vary significantly: 70GB (sh=3), 39GB (sh=2), and 27GB (sh=1). Despite the substantial storage overhead, segmentation accuracy saturates or even decreases with higher degrees. This mirrors classic results in point cloud color augmentation: beyond a certain point, extra color detail saturates accuracy gains.

**Ablation on the grid sampling method.** We further analyze the effectiveness of our proposed reweighted sampling strategies by comparing them with random sampling, as summarized in Table 7. Specifically, we explore three different reweighting criteria: *composition weight* (based on alpha compositing, measuring each Gaussian's direct contribution to the rendered image), *visibility frequency* (calculated as the number of views in which each Gaussian is visible divided by the total number of views), and a combination of both. Among these, composition-based reweighting achieves the best performance, with mIoU scores of 79.0 on ScanNet and 45.1 on ScanNet++. In contrast, visibility frequency alone yields limited improvement because occlusions frequently occur in indoor environments, causing many Gaussians to appear infrequently or inconsistently across views. Consequently, visibility frequency alone does not reliably reflect the true importance of each Gaussian for accurate semantic segmentation. The results highlight the importance of prioritizing

Table 6: **Ablation study on additional features and sampling methods.** This table reports semantic segmentation performance (mIoU) on ScanNet Dai et al. (2017) and ScanNet++ Yeshwanth et al. (2023) at grid resolutions of 0.02 m and 0.04 m. Rows I–V show the effect of incrementally adding features (coordinate, color, normal, scale, rotation and SH coefficients ) using random sampling. Row VI adopts our reweighted grid sampling.

| No. | Point Features | | | Gaussian Features | | | Sampling | ScanNet | | ScanNet++ | |
|---|---|---|---|---|---|---|---|---|---|---|---|
| | Coord | Color | Normal | Scale | Rotation | SH | | grid=0.02m | grid=0.04m | grid=0.02m | grid=0.04m |
| I | ✓ | | | | | | Random | 62.7 | 60.7 | 32.5 | 30.5 |
| II | ✓ | ✓ | | | | | Random | 74.6 | 73.8 | 37.6 | 36.4 |
| III | ✓ | ✓ | ✓ | | | | Random | 77.5 | 76.6 | 42.2 | 41.0 |
| IV | ✓ | ✓ | ✓ | ✓ | ✓ | | Random | 78.4 | 77.5 | 43.7 | 42.7 |
| V | ✓ | ✓ | ✓ | ✓ | ✓ | ✓ | Random | 78.6 | 78.1 | 44.3 | 43.5 |
| VI | ✓ | ✓ | ✓ | ✓ | ✓ | ✓ | Reweighted | **79.0** | 78.7 | **45.1** | 44.7 |

Table 7: **Ablation on grid sampling method and reweighting factors.** We compare random sampling with reweighted sampling using different reweighting strategies, and report the mIoU scores on ScanNet++ and ScanNet.

| Sampling method | Reweighted factor | Scannet | Scannet++ |
|---|---|---|---|
| Random | - | 78.6 | 44.3 |
| Reweighted | Visibility Frequency | 78.3 | 44.7 |
| Reweighted | Composition Weight | **79.0** | **45.1** |
| Reweighted | Mixture | 78.9 | 44.9 |

Table 8: **Ablation on backbone networks.** We compare the performance of our Gaussian-based method (GCept) integrated into two distinct network architectures (SparseUNet Graham et al. (2018) and PTv3 Wu et al. (2024)) against their original implementations.

| Backbone | Method | Scannet++ | | | Scannet | | |
|---|---|---|---|---|---|---|---|
| | | OA | mAcc | mIou | OA | mAcc | mIou |
| SpConv | Original | 87.8 | 46.4 | 35.8 | 90.7 | 81.8 | 73.4 |
| | GCept | 88.2 | 49.3 | 38.4 | 91.3 | 83.4 | 75.2 |
| PTv3 | Original | 88.7 | 54.7 | 42.6 | 91.8 | 85.3 | 77.5 |
| | GCept | 89.4 | 56.8 | 45.1 | 92.5 | 86.2 | 79.0 |

Gaussians based on their meaningful visual and geometric contributions, as captured effectively by our composition-based reweighting strategy.

**Ablation on backbone networks.** We evaluate the flexibility of our method by integrating it into two widely used network backbones: SparseUNet (SpConv Graham et al. (2018)) and PTv3 Wu et al. (2024). As shown in Table 8, GCept consistently improves performance across both backbones. Specifically, compared to the original SparseUNet baseline, GCept yields significant improvements of 2.6% mIoU on ScanNet++ and 1.8% mIoU on ScanNet. Similar gains are observed for the stronger PTv3 backbone, with improvements of 2.5% and 1.5% mIoU on ScanNet++ and ScanNet, respectively. These results demonstrate that our approach is general and can be seamlessly incorporated into different network architectures, further validating its broad applicability in 3D semantic segmentation.

## 5 CONCLUSION

We have proposed a framework that bridges 3D reconstruction and perception by employing Gaussian Splatting to transform raw sensor data into a continuous Gaussian-based representation. By grouping spatially proximate points into 3D Gaussians, our approach creates a compact density field that preserves geometric details often lost in conventional approaches. Evaluations across multiple indoor benchmarks demonstrate superior performance in semantic and instance segmentation tasks. Comprehensive ablation studies substantiate the efficacy of our Gaussian-based representation and adaptive sampling in enhancing input fidelity for downstream 3D perception tasks.

**Limitations and future work.** Although our GCept exhibits considerable potential in augmenting 3D perception tasks, its evaluation has thus far been confined to indoor environments. Extending our framework to outdoor scenes Behley et al. (2019); Sun et al. (2020); Fong et al. (2022); Caesar et al. (2020) presents additional challenges, including dynamic objects, variable illumination conditions, and expansive scenes. In future investigations, we intend to address these constraints by incorporating temporal consistency mechanisms for mobile objects, developing adaptive feature extraction methodologies to accommodate fluctuating lighting conditions, and designing efficient hierarchical processing strategies for expansive scenes.

## ETHICS STATEMENT

All authors of this paper have read and adhered to the ICLR Code of Ethics. Our work focuses on improving 3D perception through Gaussian Splatting representations, which is a technical contribution to computer vision and machine learning. The methods presented do not involve human subjects, do not raise concerns about harmful applications, and follow standard practices in academic research. The datasets used in our experiments are publicly available benchmarks commonly used in the research community. We do not foresee any ethical concerns arising from this work, and we are committed to responsible research practices.

## REPRODUCIBILITY STATEMENT

We are committed to ensuring the reproducibility of our work. The paper provides comprehensive implementation details in Section 3.1, including network architectures, training procedures, and hyperparameters. Detailed experimental settings are described in Section 4, including dataset preprocessing steps, evaluation metrics, and comparison protocols. Complete ablation studies with specific parameter settings are provided in the main paper and supplementary materials. We plan to release our source code, trained models, and detailed implementation instructions upon acceptance to facilitate reproduction of our results. All experiments were conducted using publicly available datasets with standard evaluation protocols to ensure fair comparison with existing methods.

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

# A    ADDITIONAL IMPLEMENTATION DETAILS

This section provides a comprehensive overview of the implementation details for our proposed framework, GCept. We describe the methodology for initializing Gaussian splatting parameters, explaining how spatially clustered points are represented as 3D Gaussians with attributes such as mean, scale, and rotation. Furthermore, we explain the configuration of the Point Transformer v3 (PTv3) backbone, encompassing training protocols, loss functions, and hardware specifications.

## A.1    GAUSSIAN SPLATTING PARAMETERS

**Initialization of Gaussian parameters**    Gaussian splattingKerbl et al. (2023) constitutes a fundamental component of our methodology, wherein spatially clustered points are consolidated into 3D Gaussians. Each Gaussian $\mathcal{G}_j$ is characterized by its mean $\mu_j$, scale matrix $S_j$, and rotation matrix $R_j$. These parameters are formulated as follows:

- **Mean ($\mu_j$):** Computed as the centroid of the points within the cluster:

$$\mu_j = \frac{1}{|\mathcal{G}_j|} \sum_{p_k \in \mathcal{G}_j} p_k,$$

    where $p_k$ denotes the 3D coordinates of each constituent point.
- **Scale ($S_j$):** Derived from the eigenvalues of the covariance matrix of the clustered points, quantifying the dispersion along the principal axes.
- **Rotation ($R_j$):** Established utilizing the eigenvectors of the covariance matrix, determining the orientation of the Gaussian.

For appearance modeling, spherical harmonics coefficients are computed to encode view-dependent chromatic variations, yielding a concise yet expressive representation of the scene's visual characteristics.

## A.2    PTV3 MODEL SETTINGS

Our implementation utilizes the Point Transformer v3 (PTv3) architecture Wu et al. (2024) as the principal backbone for 3D perception tasks. The training configurations are enumerated in Table 9. We conduct model training ab initio, employing a consistent set of parameters across diverse indoor datasets, including ScanNet, ScanNet++, and S3DIS. This homogeneous approach serves to standardize the training regimen, facilitating equitable comparisons and reproducibility across various benchmarks.

The training protocol incorporates a warm-up period spanning 40 epochs, permitting the model to achieve stability before the learning rate commences its decay. We extend training to a total of 800 epochs to ensure thorough convergence. The optimization criterion employed is CrossEntropy, deemed appropriate for multi-class semantic segmentation objectives.

Table 9: **Semantic and Instance Segmentation Configuration Parameters.**

| Settings | Value | |
|---|---|---|
|  | Semantic | Instance |
| optimizer | AdamW | AdamW |
| scheduler | Cosine | Cosine |
| criteria | CrossEntropy | CrossEntropy |
| learning rate | 5e-3 | 5e-3 |
| block lr scaler | 0.1 | 0.1 |
| weight decay | 5e-2 | 5e-2 |
| batch size | 12 | 12 |
| warmup epochs | 40 | 40 |
| epochs | 800 | 800 |

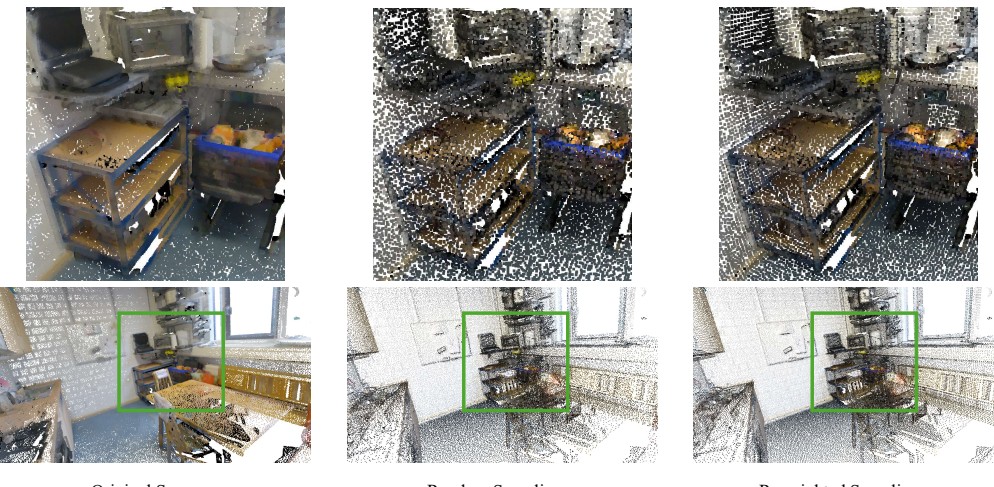

| Original Scene | Random Sampling | Reweighted Sampling |

Figure 4: Visualization of our reweighted sampling methodology. From left to right: the original scene, random sampling, and reweighted sampling. The reweighted approach preserves substantially more intricate details, particularly around object boundaries, compared to random sampling.

Table 10: Per-class performance comparative analysis on ScanNet dataset

| Method | Metric | Mean | wall | floor | cabi | bed | chair | sofa | table | door | window | bkshf | pict | counter | desk | curt | fridge | sh curt | toilet | sink | bath | oth fur |
|---|---|---|---|---|---|---|---|---|---|---|---|---|---|---|---|---|---|---|---|---|---|---|
| SpCov | IoU | 69.3 | 84.9 | 95.8 | 54.8 | 79.7 | 78.8 | 61.5 | 71.5 | 65.5 | 57.8 | 78.0 | 31.3 | 60.4 | 68.2 | 75.5 | 60.9 | 73.9 | 92.1 | 62.0 | 83.8 | 50.2 |
|  | Acc | 78.7 | 89.2 | 95.2 | 78.5 | 84.8 | 77.2 | 89.1 | 80.1 | 78.9 | 64.4 | 85.7 | 55.4 | 83.5 | 82.9 | 80.4 | 60.2 | 79.8 | 93.7 | 73.9 | 91.0 | 51.2 |
| MinkNet | IoU | 72.2 | 85.3 | 96.4 | 61.6 | 79.6 | 90.4 | 82.9 | 69.9 | 66.0 | 63.0 | 82.2 | 39.5 | 65.4 | 62.5 | 71.5 | 64.3 | 60.0 | 94.6 | 67.5 | 86.6 | 55.2 |
|  | Acc | 80.4 | 88.7 | 95.8 | 82.5 | 81.5 | 92.4 | 90.4 | 70.8 | 86.2 | 65.3 | 93.6 | 52.9 | 79.1 | 91.8 | 87.9 | 72.0 | 62.4 | 94.2 | 72.1 | 88.8 | 60.4 |
| ST | IoU | 74.3 | 87.1 | 96.6 | 65.0 | 81.2 | 89.7 | 76.8 | 76.3 | 69.1 | 66.3 | 79.9 | 35.7 | 64.9 | 69.2 | 73.3 | 66.4 | 73.9 | 94.2 | 70.9 | 88.5 | 60.6 |
|  | Acc | 83.3 | 92.2 | 97.3 | 74.3 | 85.7 | 92.4 | 93.2 | 83.7 | 80.8 |  | 89.1 | 53.5 | 84.2 | 82.7 | 83.7 | 74.9 | 79.9 | 96.2 | 78.3 | 91.7 | 65.6 |
| PointNeXt | IoU | 71.5 | 86.1 | 95.5 | 64.8 | 80.6 | 90.0 | 80.8 | 66.9 | 64.8 | 65.5 | 76.9 | 39.1 | 63.6 | 62.7 | 73.1 | 60.9 | 66.2 | 92.9 | 55.8 | 85.1 | 58.7 |
|  | Acc | 82.0 | 93.3 | 97.4 | 80.5 | **88.0** | 94.6 | 87.3 | 75.8 | 70.0 | **85.8** | **95.4** | 44.3 | 80.7 | 87.1 | 83.7 | 76.8 | 71.8 | 95.1 | 71.6 | 91.2 | 69.4 |
| OctFormer | IoU | 75.7 | 87.6 | 96.5 | 70.3 | 82.2 | 91.5 | 86.7 | 74.0 | 69.5 | 68.2 | 81.7 | 39.7 | 67.4 | 67.2 | 76.3 | 68.5 | 66.3 | 93.5 | 70.5 | **90.0** | 65.8 |
|  | Acc | 83.2 | 96.6 | 98.5 | 82.2 | 87.2 | 95.5 | 91.2 | 79.6 | 77.0 | 80.5 | 92.9 | 46.2 | 78.9 | **92.6** | 81.1 | 73.0 | 69.7 | 96.1 | 81.2 | 93.5 | 71.3 |
| Swin3D | IoU | 76.4 | 88.5 | 96.4 | 68.8 | 82.2 | 92.0 | 86.2 | 77.5 | 73.0 | 72.9 | 79.2 | 42.0 | 63.8 | 69.8 | 78.2 | 63.5 | **77.1** | 94.9 | 68.2 | 85.9 | 67.8 |
|  | Acc | **86.3** | 94.2 | 98.1 | 80.8 | 87.0 | **97.3** | 93.2 | 84.3 | **88.5** | 83.8 | 90.7 | 54.9 | 84.0 | 89.0 | 86.8 | 81.1 | **85.2** | 98.2 | 81.2 | **94.8** | 73.8 |
| PTv1 | IoU | 70.6 | 86.2 | **97.1** | 56.1 | 81.0 | 80.1 | 62.8 | 72.8 | 66.7 | 59.1 | 79.3 | 32.6 | 61.7 | 69.4 | 76.8 | 62.2 | 75.2 | 93.4 | 63.3 | 85.0 | 51.5 |
|  | Acc | 78.5 | 88.9 | 94.9 | 78.3 | 84.6 | 77.0 | 88.9 | 79.8 | 78.7 | 64.1 | 85.4 | **55.1** | 83.3 | 82.7 | 80.1 | 59.9 | 79.5 | 93.4 | 73.6 | 90.7 | 50.9 |
| PTv2 | IoU | 75.4 | 87.6 | 96.4 | 65.4 | 83.4 | 90.9 | 84.2 | 74.1 | 71.5 | 68.6 | 77.4 | **44.2** | 66.4 | 72.3 | 73.4 | 69.5 | 73.2 | 93.1 | 67.5 | 87.9 | 61.6 |
|  | Acc | 84.5 | 94.9 | 98.6 | 80.2 | 87.4 | 96.3 | 90.3 | 82.7 | 82.5 | 79.3 | 91.2 | 52.1 | **85.0** | 86.7 | 85.2 | 79.7 | 78.9 | **98.3** | 80.6 | 92.3 | 67.4 |
| PTv3 | IoU | 77.5 | **88.7** | 96.4 | 71.0 | 82.9 | 92.9 | 85.1 | 79.7 | 72.7 | 70.6 | 83.8 | 39.5 | 69.4 | **74.6** | 78.2 | 74.3 | 69.4 | 94.0 | 69.3 | 89.8 | 67.7 |
|  | Acc | 85.3 | **96.4** | **98.9** | **85.2** | 87.4 | 96.8 | 93.3 | **87.5** | 80.0 | 82.2 | 93.1 | 50.7 | 81.4 | 87.3 | **88.3** | 78.7 | 74.2 | 98.1 | 79.8 | 92.6 | **73.9** |
| GCept | IoU | **79.0** | **88.7** | 96.0 | **73.7** | **83.1** | **93.8** | **87.9** | **79.8** | **75.4** | **74.5** | **85.6** | 41.1 | **70.3** | 73.9 | **78.9** | **77.4** | 75.0 | **96.0** | **70.7** | 89.2 | **67.9** |
|  | Acc | 86.2 | 96.1 | 98.5 | 82.7 | 87.3 | 97.2 | 93.9 | 86.1 | 84.0 | **85.8** | 94.2 | 50.6 | 83.1 | 90.1 | 86.7 | **83.0** | 79.7 | **98.3** | **82.0** | 92.2 | 73.2 |

# B  VISUALIZATION OF REWEIGHTED SAMPLING

In this section, we illustrate the efficacy of our proposed reweighted sampling methodology utilizing a point cloud visualization approach. For a given point cloud within a scene, we apply Gaussian splatting to compute an importance metric for each point, wherein each point is represented as a Gaussian center with associated parameters. These importance metrics are derived from alpha compositing contributions and reflect each Gaussian's prominence in the final rendered image.

To explain the effect of our sampling strategy, we execute grid subsampling on a fixed grid of 0.02 m employing two distinct methodologies: random sampling and our proposed reweighted sampling. In the random sampling scenario, points are selected arbitrarily from each grid cell without consideration of their importance. Conversely, our reweighted sampling methodology selects points probabilistically, with elevated likelihood assigned to those Gaussians that exhibit more substantial alpha compositing contributions.

In Figure 4, we juxtapose the outcomes of random sampling against our reweighted sampling approach on an authentic indoor scene. The original point cloud (left) captures the intricate layout of the environment, encompassing various objects and structures. Random sampling (middle) discards numerous fine details, resulting in a sparser and less informative representation. In contrast, our

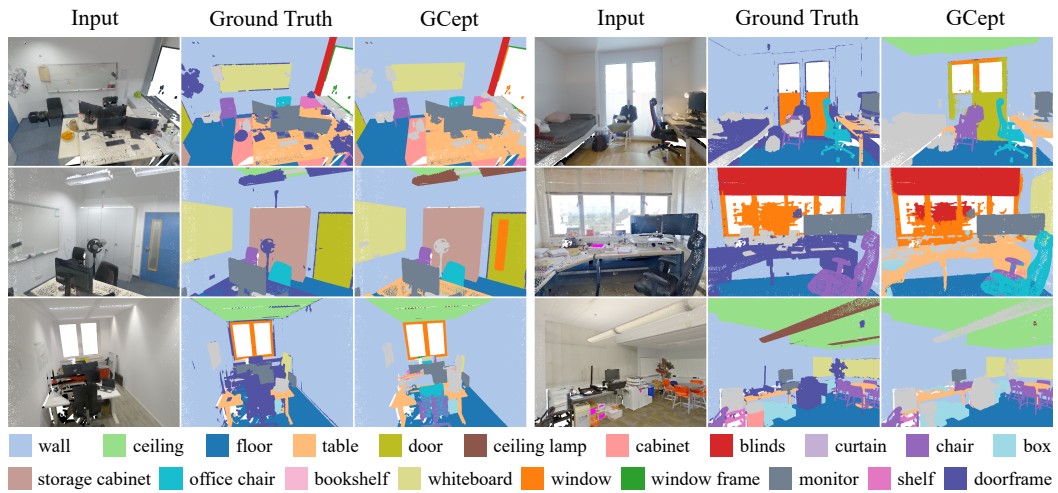

| Input | Ground Truth | GCept | Input | Ground Truth | GCept |

wall    ceiling    floor    table    door    ceiling lamp    cabinet    blinds    curtain    chair    box

storage cabinet    office chair    bookshelf    whiteboard    window    window frame    monitor    shelf    doorframe

Figure 5: **Visualization of semantic segmentation results on ScanNet++ Yeshwanth et al. (2023).**

reweighted sampling (right) retains Gaussians with higher alpha compositing contributions, thereby preserving critical geometric and appearance attributes and yielding a more faithful depiction of the scene.

## C COMPREHENSIVE SEMANTIC SEGMENTATION RESULTS

### C.1 PER-CLASS METRICS ANALYSIS

Table 10 presents the per-class Intersection-over-Union (IoU) and accuracy metrics for our GCept methodology compared to extant baselines on the ScanNet Dai et al. (2017) benchmark. Each column corresponds to a specific semantic category, with the second column reporting the mean IoU and accuracy across all classes. GCept consistently surpasses other approaches, particularly in categories exhibiting complex geometry or nuanced appearance cues (e.g., "refrigerator", "curtain"), demonstrating the advantages of our Gaussian-based representation and reweighted sampling strategy.

### C.2 QUALITATIVE RESULTS

The qualitative outcomes of point cloud semantic segmentation are depicted in Figure 5, wherein our GCept model generates predictions that closely correspond with the ground-truth annotations. Notably, GCept effectively captures fine-grained structural intricacies and delivers precise predictions in complex indoor environments. For instance, in ScanNet++ scenes with challenging object arrangements, GCept accurately delineates objects with intricate boundaries, such as differentiating furniture from walls and identifying diminutive objects within cluttered settings. These outcomes underscore the robustness and precision of GCept in addressing the complexities of 3D semantic segmentation tasks. For supplementary visualizations and additional experimental particulars, please refer to our supplementary materials.

To further elucidate the qualitative enhancements offered by our methodology, Figure 6 presents additional segmentation outcomes on challenging indoor scenes from ScanNet. Each sub-figure contrasts our GCept predictions with the ground truth annotations. It is noteworthy that GCept more accurately delineates object boundaries and consistently manages small, cluttered regions (e.g., the extremities of tables and chairs), thereby diminishing misclassifications compared to baseline methodologies.

These qualitative exemplars corroborate the quantitative advancements observed in Table 10, emphasizing that the rich geometric and appearance attributes captured by GCept facilitate more precise segmentation across a diverse array of object categories.

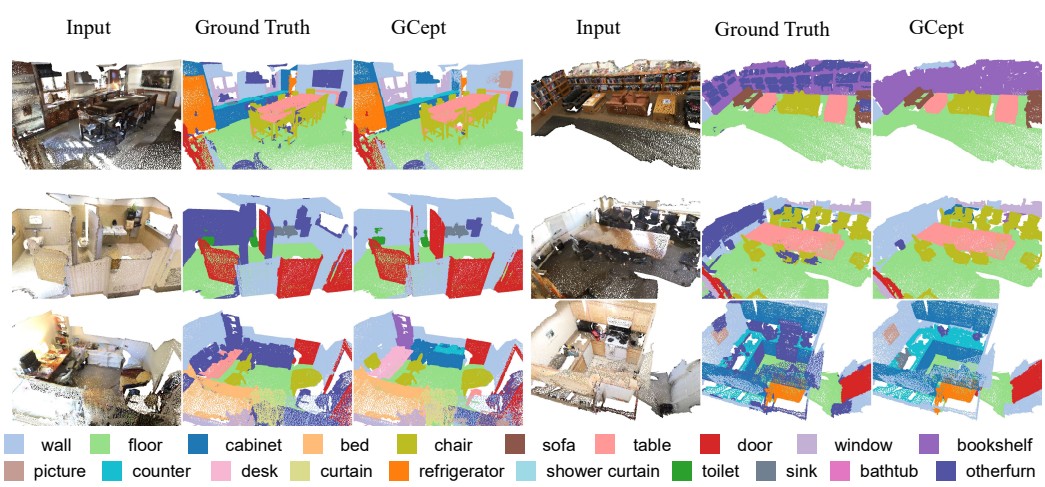

Figure 6: Visualization of semantic segmentation results on ScanNet Dai et al. (2017).

# D COMPARATIVE ANALYSIS OF GAUSSIAN SPLATTING IMPLEMENTATIONS

To demonstrate the versatility and robustness of our proposed framework, we conducted comparative experiments utilizing different Gaussian Splatting implementations. The objective of this analysis is to ascertain that the benefits of our GCept approach are not confined to a specific implementation but rather emanate from the fundamental principles of employing Gaussian primitives for 3D perception.

## D.1 IMPLEMENTATION VARIANTS

We evaluated the following Gaussian Splatting implementations:

- **3DGS-Original** Kerbl et al. (2023): The original 3D Gaussian Splatting implementation, which utilizes anisotropic 3D Gaussians with spherical harmonics for view-dependent appearance.
- **2DGS** Huang et al. (2024): A novel approach that collapses the 3D volume into a set of 2D oriented planar Gaussian disks, providing view-consistent geometry while modeling surfaces intrinsically. It employs perspective-accurate 2D splatting with ray-splat intersection and incorporates depth distortion and normal consistency terms to enhance reconstruction fidelity. This approach achieves noise-free, detailed geometry while maintaining competitive appearance quality and real-time rendering capabilities.

## D.2 PERFORMANCE COMPARATIVE ANALYSIS

Table 11 presents the semantic segmentation performance (mIoU) achieved by integrating different Gaussian Splatting implementations into our GCept framework. The experiments were conducted on ScanNet and ScanNet++ datasets.

Table 11: Performance comparative analysis of different Gaussian Splatting implementations within our GCept framework

| GS Implementation | Description | ScanNet (mIoU) | ScanNet++ (mIoU) |
| --- | --- | --- | --- |
| 3DGS Kerbl et al. (2023) | Original implementation | 78.6 | 44.7 |
| 2DGS Huang et al. (2024) | Our default | 79.0 | 45.1 |

As illustrated in Table 11, all Gaussian Splatting implementations provide substantial enhancements over conventional point-based approaches. While 2DGS yields the optimal performance and constitutes our default selection due to its advantageous balance of accuracy and efficiency, the alternative

implementations also exhibit robust performance. This substantiates that the gains from our GCept framework originate primarily from the fundamental advantages of Gaussian-based representation and reweighted sampling rather than from implementation-specific nuances.

# E    COMPUTATIONAL EFFICIENCY ANALYSIS

We present a detailed analysis of the computational aspects of our GCept framework compared to conventional point-based methods. This examination encompasses preprocessing time, training efficiency, and inference speed across diverse datasets and configurations.

## E.1    PREPROCESSING TIME

Table 12 presents the preprocessing times required for transforming raw point clouds into Gaussian-based representations across different datasets. We report the average duration per scene for each stage of the preprocessing pipeline.

Table 12: Preprocessing time for GCept (seconds per scene)

| Process | ScanNet | ScanNet++ |
|---|---|---|
| Point cloud loading | 0.5 | 0.6 |
| Gaussian initialization | 2.7 | 3.4 |
| Gaussian Splatting Optimization | 762.3 | 951.8 |
| Total GCept preprocessing | 765.5 | 955.8 |

While GCept necessitates additional preprocessing duration compared to conventional methodologies, this computational overhead constitutes a one-time cost that is amortized over multiple utilizations of the dataset. The advantages of the enhanced representation significantly outweigh this initial preprocessing investment, as evidenced by the substantial improvements in segmentation accuracy.

## E.2    TRAINING EFFICIENCY

Table 13 juxtaposes the training efficiency of our GCept framework against conventional point-based methodologies. We report the average duration per epoch and GPU memory consumption during training.

Table 13: Training efficiency comparative analysis on ScanNet

| Method | Model Params (M) | GFlops | Time/epoch (s) | GPU Memory (GB) | mIoU |
|---|---|---|---|---|---|
| PTv3(grid=0.02m) | 46.2 | 42.5 | 160.8 | 60.1 | 77.5 |
| PTv3(grid=0.04m) | 46.2 | 39.0 | 150.8 | 36.4 | 76.6 |
| DSConv-XXLZhang et al. (2024) | 25.9 | 99.6 | N/A | N/A | 77.8 |
| GCept(grid=0.02m) | 47.0 | 48.1 | 181.4 | 61.9 | 79.0 |
| GCept(grid=0.04m) | 47.0 | 45.1 | 170.9 | 40.2 | 78.7 |

Our examination reveals that GCept introduces a modest computational overhead compared to the baseline PTv3 approach. Specifically, with a grid size of 0.02m, GCept requires 1.7% more parameters (47.0M vs. 46.2M) and 13.2% higher computational cost (48.1 vs. 42.5 GFlops) than PTv3. This translates to a 12.8% increase in training duration and 3.0% higher memory utilization.

When employing a larger grid size of 0.04m, we observe similar parameter counts but a 15.6% increase in GFlops compared to the corresponding PTv3 configuration, resulting in 13.3% longer training epochs and 10.4% higher memory utilization. However, this computational investment yields significant performance enhancements: GCept surpasses PTv3 by 1.5% mIoU (79.0% vs. 77.5%) at 0.02m grid resolution and by 2.1% mIoU (78.7% vs. 76.6%) at 0.04m grid resolution.

Compared to DSConv-XXL, GCept employs more parameters but necessitates substantially lower computational complexity (48.1 vs. 99.6 GFlops) while still achieving superior segmentation perfor-

mance (+1.2% mIoU). These outcomes demonstrate that the enhanced representational capacity of our Gaussian-based features justifies the modest computational overhead, offering an advantageous balance between performance and efficiency.

### E.3 INFERENCE SPEED

Table 14 presents the inference speed for different methodologies and configurations on a single NVIDIA L40S GPU.

Table 14: Inference speed comparative analysis with latency and throughput

| Method | ScanNet | | ScanNet++ | |
|---|---|---|---|---|
| | Latency (ms) | TP (scenes/s) | Latency (ms) | TP (scenes/s) |
| PTv3 | 48.81 | 20.49 | 66.87 | 14.95 |
| GCept | 51.76 | 19.32 | 75.76 | 13.20 |

In terms of inference performance, GCept exhibits a slight increase in latency compared to the PTv3 baseline. On the ScanNet dataset, GCept's latency is 6.0% higher (51.76ms vs. 48.81ms). On the more complex ScanNet++ dataset, we observe a 13.3% increase in latency (75.76ms vs. 66.87ms).

While these differences represent a modest computational cost, it is imperative to contextualize them against the significant gains in segmentation quality: GCept delivers 1.5-2.1% higher mIoU compared to PTv3 and outperforms DSConv-XXL across all metrics. For numerous real-world applications where accuracy is paramount (e.g., robotics navigation, precise object manipulation, or detailed scene understanding), this performance-to-efficiency ratio represents a favorable compromise.

### E.4 SUMMARY

Our computational analysis demonstrates that GCept introduces a modest computational overhead compared to conventional point-based methodologies. While preprocessing necessitates approximately 13-16 minutes per scene, this constitutes a one-time cost amortized across multiple utilizations. During training, GCept increases model parameters marginally (1.7% more than PTv3) with 13-15% higher computational cost in GFlops. This translates to 12-13% longer training duration and 3-10% higher memory utilization, contingent upon grid size. At inference time, GCept is approximately 6% slower on ScanNet and 13% slower on ScanNet++.

Despite these modest computational costs, GCept achieves significant performance enhancements compared to current state-of-the-art methodologies: +1.5-2.1% mIoU over PTv3 and +1.2% mIoU over DSConv-XXL, while requiring less than half the computational complexity of DSConv-XXL. This advantageous performance-to-cost ratio renders GCept particularly attractive for applications where segmentation quality is paramount. Furthermore, as hardware capabilities continue to advance and Gaussian splatting implementations become more optimized, we anticipate the computational overhead to diminish further, rendering this approach increasingly viable for real-time applications.

## F USE OF LARGE LANGUAGE MODELS

In accordance with ICLR 2026 guidelines, we disclose the use of Large Language Models (LLMs) in the preparation of this manuscript. LLMs were employed primarily as writing assistance tools to enhance the clarity, coherence, and grammatical accuracy of the text.

All core research ideas, methodology design, experimental planning, implementation, and analysis were conducted independently by the authors. The LLMs did not contribute to research ideation, experimental design, or the interpretation of results. The technical content, including mathematical formulations, algorithmic descriptions, and scientific conclusions, represents the original work and insights of the authors. LLM assistance was limited to improving the linguistic presentation of these ideas without altering their substance or validity.

