# OpenReview forum: "Seeing Beyond Points: Adaptive Gaussian Primitives for 3D Perception"
_ICLR.cc/2026/Conference — ICLR 2026 Conference Withdrawn Submission_

### Official Review · Reviewer_uG6u · 2025-10-21

**Soundness:** 3
**Presentation:** 3
**Contribution:** 3
**Rating:** 4
**Confidence:** 5

**Summary:**

GCept is a unified 3D perception framework designed to overcome the limitations of raw point clouds, which struggle to capture continuous geometry and fine visual details. It transforms discrete points into adaptive 3D Gaussian primitives with optimized covariances and spherical harmonics encoding, forming a smooth and information-rich density field. Using an alpha-guided sampling mechanism derived from Gaussian Splatting, GCept selectively retains the most informative primitives, enhancing geometric and appearance fidelity. This enriched representation integrates seamlessly into standard 3D perception networks, achieving state-of-the-art results in semantic and instance segmentation across multiple benchmarks including ScanNet, ScanNet++, ScanNet200, and S3DIS—effectively bridging the gap between 3D reconstruction and perception.

**Strengths:**

[S1] The paper effectively showcases the potential of novel 3D representations, particularly 3D Gaussian Splatting (3D-GS), in capturing fine-grained scene geometry. Unlike conventional point-based methods that suffer from discontinuity, 3D-GS provides a continuous and differentiable representation of spatial structures. This demonstrates how adopting 3D-GS as a foundational representation can lead to a more faithful reconstruction of underlying geometries in complex real-world scenes.

[S2] The authors present a clear and convincing motivation for reweighting point samples using the alpha attributes of 3D-GS. They explicitly connect this design choice to the goal of emphasizing geometrically and visually informative regions, addressing the inherent limitations of uniform sampling in point clouds. This rationale provides readers with a strong understanding of why leveraging 3D-GS properties leads to improved spatial awareness and better feature aggregation.

[S3] The consistent performance improvement over standard 3D-GS baselines indicates the general applicability of their proposed strategy. It suggests that their method can serve as a plug-and-play enhancement for existing 3D perception frameworks, particularly those based on point or voxel representations. Overall, the results demonstrate that the proposed approach is not only effective on its own but also scalable to more advanced 3D point cloud architectures.

**Weaknesses:**

**[W1] Missing Recent Benchmarks.**

Although the authors claim that their model achieves state-of-the-art performance, they omit several recent baselines that outperform PTv3. For instance, *Sonata* [1] and *DIR* [2] respectively achieve mIoU scores of 79.2 and 80.5 on ScanNet, surpassing GCept. Nevertheless, I acknowledge that GCept outperforms Sonata on S3DIS, ScanNet++, and ScanNet200. Since [1] and [2] were published in March, they should not be considered concurrent works, and thus should be included in the comparison. Furthermore, the authors are encouraged to moderate their claims regarding state-of-the-art achievement.

**[W2] Model Design.**

The proposed method introduces a novel voxelization strategy based on the alpha attribute of 3D-GS. However, the alpha value varies with the viewing direction, implying that it does not consistently represent point density. For instance, the compositing weight depends on transmittance—an accumulated quantity during alpha blending—thus yielding inconsistent values across viewpoints. This raises concerns about whether alpha-based reweighting is a valid design choice. Consequently, the benefit of reweighted sampling appears marginal.

Additionally, the design overlooks the scale of each Gaussian, as the weighting function is independent of it. Large Gaussians should influence multiple voxels, yet the current approach treats 3D-GS as discrete points rather than continuous fields. This contradicts their own statement in the Introduction: *“We view 3D Gaussians not as substitutes for points, but as the next logical step in their evolution, following the lineage from raw coordinates to surfels and attribute-augmented points”* (L40–43).

**[W3] Adding DIR in Table 2.**

The authors should include DIR [2] in Table 2 and revise their claim to state that GCept slightly underperforms DIR.

**[W4] Limited Application.**

As shown in Table 12, the 3D-GS optimization process imposes a significant computational burden, requiring long preprocessing times before the model can be utilized. Furthermore, evaluations are limited to indoor datasets. Although the authors briefly mention this in the limitation section, 3D-GS techniques are also effective for outdoor datasets. Therefore, additional experiments on outdoor scenes would enhance the comprehensiveness of the evaluation.

**[W5] Incomplete Writing.**

Several segments in the manuscript remain unfinished. While these do not substantially hinder readability, they should be addressed during the rebuttal phase.

- Missing citations (L103–107).

[1] **Sonata:** *Self-supervised Learning of Reliable Point Representations*, Wu et al.

[2] **DINO in the Room:** *Leveraging 2D Foundation Models for 3D Segmentation*.

**Questions:**

No questions to the authors.

---

### Official Review · Reviewer_26DD · 2025-10-23

**Soundness:** 3
**Presentation:** 3
**Contribution:** 2
**Rating:** 6
**Confidence:** 4

**Summary:**

The paper introduces GCept, a unified framework that transforms sparse point clouds into adaptive Gaussian primitives. Instead of treating points as discrete samples, GCept clusters spatially neighboring points into anisotropic 3D Gaussians equipped with covariance, rotation, and spherical harmonics to encode both geometry and view-dependent color. It further proposes an alpha-guided reweighted sampling strategy derived from Gaussian Splatting compositing weights to prioritize visually and geometrically informative regions. By integrating into standard 3D backbones such as Point Transformer v3 and SparseUNet, GCept achieves SOTA performance across benchmarks including ScanNet, ScanNet++, ScanNet200, and S3DIS.

**Strengths:**

1. GCept advances 3D perception by reinterpreting point clouds as continuous Gaussian fields that capture geometry, uncertainty, and view-dependent color.
2. The proposed reweighted sampling mechanism uses compositing weights from Gaussian Splatting to retain only the most informative primitives, to improve accuracy around object boundaries and fine structures.
3. By integrating into standard 3D backbones such as Point Transformer v3 and SparseUNet, GCept achieves SOTA performance across benchmarks, including ScanNet, ScanNet++, ScanNet200, and S3DIS.

**Weaknesses:**

1. Current evaluations are confined to indoor datasets; the framework’s performance in outdoor or dynamic environments remains untested.
2. The method seems interesting, but I am less enthusiastic about the problem (point cloud segmentation), which makes me feel less excited about the paper and its contribution. In terms of this problem (point cloud segmentation), I think the more impactful problem is not about keeping pushing the performance on the current benchmark, but the one that can push the frontier of the research community and industry, like open-vocabulary segmentation, large-scale (pre) training, PEFT adaptation of pre-trained foundational model, domain generalization, etc. In that case, I feel the contribution of this paper is slightly limited.

**Questions:**

**Major:**
1. Is the raw input of the method (the sensory input) RGB + Depth or RGB + Point Cloud (in world coordinates)? If I understand correctly, the Gaussian is not the raw input but some intermediate representation after processing the raw input. In that case, the Input in Table 1 shall not be Gaussian.

**Minor:**
1. Can GCept be extended to handle temporal changes and moving objects in outdoor environments such as autonomous driving datasets (e.g., nuScenes, KITTI)?
2. How might future work reduce the computational overhead of Gaussian Splatting while preserving its representational richness?
3. Could GCept be fused with language or vision-language models (e.g., CLIP or 3D-LLMs) for open-vocabulary segmentation?

---

### Official Review · Reviewer_wbE4 · 2025-10-31

**Soundness:** 2
**Presentation:** 3
**Contribution:** 2
**Rating:** 4
**Confidence:** 5

**Summary:**

This paper proposes a unified 3D perception framework that evolves raw points into adaptive Gaussian primitives, representing a natural progression in point cloud enrichment. It  groups spatially proximate points into 3D Gaussians with optimized covariances and spherical harmonics encoding, forming a continuous density field that preserves intricate geometric structures and subtle visual details often lost in traditional pipelines.

**Strengths:**

1. The structure of this paper is clear and well-organized.
2. The experimental results verify the effectiveness of the proposed method to some extent.

**Weaknesses:**

1. The research motivation is unreasonable. Gaussian Splatting is an independent research field. Although it can promote 3D perception, simply transferring it to this task is more like an engineering project than a research project. What potential problems might be encountered during the transfer? If there are no obvious problems, then this paper is more like an integration than an independent research proposition.
2. The proposed method lacks novelty. [1] also introduces Gaussian Splatting to 3D point clouds, which is almost identical to the method in this paper in its core.
3. The experiments are unconvincing. This method uses Gaussian Splatting, which has not been adopted by other works. Therefore, it is unclear whether the performance improvement is due to Gaussian Splatting or the method designed in this paper. Secondly, in Table 3, the authors used spherical harmonics, while the comparison method only used positional information, which is an unfair comparison. Therefore, it is impossible to judge the effectiveness of the proposed method.
[1] Mitigating Ambiguities in 3D Classification with Gaussian Splatting

**Questions:**

See the comments below.

---

### Official Review · Reviewer_62Xx · 2025-10-31

**Soundness:** 3
**Presentation:** 3
**Contribution:** 2
**Rating:** 4
**Confidence:** 3

**Summary:**

This paper presents a framework, namely GCept, that leverages 3D Gaussian Splatting to create enriched point cloud representations for 3D perception tasks. The core idea is to transform discrete points into continuous Gaussian primitives with geometric (covariance) and appearance (SHs) information, combined with an alpha-guided sampling strategy. The work demonstrates consistent improvements across semantic and instance segmentation benchmarks. While the motivation is sound and the results are strong, the key core contribution essentially applies existing techniques without significant innovation.

**Strengths:**

In Figure 1A, the conceptual framing shows the parallel evolution of rendering and perception from raw points to Gaussians, which is compelling. The paper is well-written and easy to follow.

The approach integrates seamlessly into existing 3D perception pipelines without requiring architectural changes to backbone networks.

In terms of the quantitative performance, this method shows consistent improvements across multiple benchmarks, inc., ScanNet, ScanNet++, ScanNet200, and S3DIS.

**Weaknesses:**

The core contribution essentially applies existing techniques.

The Gaussian Representation in Section 3.2  is standard 3D Gaussian Splatting with fixed centres. The clustering of nearby points into Gaussians is straightforward (mean, PCA-based covariance).  Alpha-guided sampling is a relatively simple application of compositing weights already computed by Gaussian Splatting. The reweighting factor is just aggregating existing alpha values across views and using them for importance sampling. This is intuitive but not technically sophisticated.

Where is the learning? The Gaussians are fixed after initial optimisation, and the sampling weights are deterministic. There's no end-to-end training, no learned importance metrics, no adaptation of Gaussians for perception. Why not learn to optimise Gaussian parameters jointly with the perception task? This would be a genuine contribution rather than a two-stage approach.

From Table 12, the preprocessing takes 762-951 seconds per scene for Gaussian Splatting optimisation. The training and inference speed is also 10% slower, memory takes 3-10% higher. For a 1.5-2.5% mIoU improvement, this overhead is concerning.

The paper claims Gaussians as "the next evolution" of points, but doesn't compare with other enrichment strategies, e.g., methods like PointContrast, PPKT, or other self-supervised approaches that learn rich features per point. How does this compare to Gaussian features? Or even simpler: what if you just add colour from the RGB images to points? (No Gaussian Splatting), Or add more geometric features computed from local neighbourhoods? Or query MLPs (like in NeRF) at point locations to get learned embeddings. These enrichment strategies do not require multi-view optimisation even.

**Questions:**

In section 4.2, "we set the learning rate for the Gaussian center positions µj to zero, thereby fixing the centers of each Gaussian to their initialized positions". Why? This seems arbitrary and unexplained. For perception, wouldn't adapting positions to task-relevant features be beneficial?

In section 3.2: "we group spatially proximate points and represent each cluster as a 3D Gaussian. ", How? K-means? Grid-based? Voxelization? How is the number of Gaussians M determined relative to points N?

---

### Note · Authors · 2025-11-13

I have read and agree with the venue's withdrawal policy on behalf of myself and my co-authors.